# Core-Shell, Critical-Temperature-Suppressed V Alloy-Pd Alloy Hydrides for Hydrogen Storage—A Technical Evaluation

**DOI:** 10.3390/molecules28073024

**Published:** 2023-03-28

**Authors:** Krystina E. Lamb, Colin J. Webb

**Affiliations:** 1Australian Nuclear Science and Technology Organisation, 800 Blackburn Road, Clayton, VIC 3168, Australia; 2Monash University Affiliate, Monash University, Wellington Rd, Clayton, VIC 3800, Australia; 3Queensland Micro and Nanotechnology Centre, Griffith University, Nathan, QLD 4111, Australia

**Keywords:** palladium, vanadium, hydrogen storage, alloys

## Abstract

Hydrogen storage for energy applications is of significant interest to researchers seeking to enable a transition to lower-pollution energy systems. Two of the key drawbacks of using hydrogen for energy storage are the low gas-phase storage density and the high energy cost of the gas-phase compression. Metal hydride materials have the potential to increase hydrogen storage density and decrease the energy cost of compression by storing the hydrogen as a solid solution. In this article, the technical viability of core-shell V_90_Al_10_-Pd_80_Ag_20_ as a hydrogen storage material is discussed. LaNi_5_, LaNi_5_/acrylonitrile-butadiene-styrene copolymer mixtures, core-shell V-Pd, and core-shell V_90_Al_10_-Pd_80_Ag_20_ are directly compared in terms of reversible hydrogen-storage content by weight and volume. The kinetic information for each of the materials is also compared; however, this work highlights missing information that would enable computational dynamics modelling. Results of this technical evaluation show that V_90_Al_10_-Pd_80_Ag_20_ has the potential to increase gravimetric and volumetric hydrogen capacity by 1.4 times compared to LaNi_5_/acrylonitrile-butadiene-styrene copolymer mixtures. In addition, the literature shows that Pd_80_Ag_20_ and V_90_Al_10_ both have similarly good hydrogen permeabilities, thermal conductivities, and specific heats. In summary, this evaluation demonstrates that core-shell V_90_Al_10_-Pd_80_Ag_20_ could be an excellent, less-expensive hydrogen storage material with the advantages of improved storage capacity, handleability, and safety compared to current AB_5_-polymer mixtures.

## 1. Introduction

Intermetallic and solid hydrides have been proposed as hydrogen storage solutions instead of high-pressure gaseous hydrogen, having improved storage density and duration at lower pressures. LaNi_5_ is a well-studied representative of the AB_5_ metal alloys for hydrogen storage and has also been studied for other commercial and domestic applications, including heat pumps [1] and high-pressure actuators [2].

While there are many potential metal alloys that can absorb hydrogen under a variety of conditions, every material explored has all or many of the following problems that occur with use: decrepitation, sintering, passivation, hysteresis, low effective thermal conductivity, and high chemical reactivity.

Decrepitation is the process wherein metal particles break into smaller particles, effectively being pulverized into a fine powder. One of the causes of decrepitation is the cycling of the metal hydride from a high hydrogen to low hydrogen concentration at a temperature at which different phases form at different hydrogen concentrations [3]. This is because the difference in unit cell size and shape between different phases causes ‘fault’ lines where the particles break apart from other forces, such as gravity and thermal and physical expansion [3]. These effects pulverize the alloys during repeated cycling and result in both benefits and drawbacks. The benefits include a larger surface area to interior ratio and increasing the speed of hydrogen ad- and absorption [3]. The drawbacks include increased difficulty in handling, increased poisoning effects of contamination, and reduced heat kinetics due to poor particle packing and gaps between the particles, thereby decreasing the effective thermal conductivity [3]. Sintering can also occur in some alloys, where rather than fragmenting over temperature and pressure cycling, the particles join into larger crystals. This has the same drawbacks and benefits as decrepitation, just reversed.

Hysteresis in metal hydrides is where it takes a higher pressure to absorb hydrogen into the material than when it is released. This increases the energy cost of storage of hydrogen and the pressure rating requirements. There are several factors that contribute to hysteresis; however, the requirement for dislocations to be performed and propagated in the metal’s crystal structure with the addition and subtraction of hydrogen is regarded as a major contributing factor [4].

The effects of hysteresis and decrepitation combine to lead to low effective thermal conductivity in metal hydride beds, especially as the material creates inter-particle gaps upon decrepitation, leading to gas-phase convection and gas conduction providing most of the thermal redistribution, rather than conduction within the metal. This increases the time taken to heat and cool the metal hydride bed, decreasing the efficiency.

Other than tank design incorporating thermal management devices at the system level, there have been several methods proposed in the literature to address these issues at the material level. One is coating or embedding the metal hydrides in hydrogen-permeable materials such as polymers [5]. This offers the possibility of improving the effective thermal conductivity and handleability, and reducing the deactivation/poisoning and the chance of run-away reactions in the case of accidental air exposure [5]. Another method proposed is the modification of the metal hydrides to change the critical temperature, which is the temperature whereat further cooling results in a phase change in the alloy, to enable phase-transition free operation at convenient temperatures [6].

Suppression of the critical temperature of the hydride phase for alloys has been of interest in metal hydride research for the purpose of producing efficient and cheaper solid-state hydrogen-purification membranes [6]. Vanadium, palladium, tantalum, niobium, and zirconium, are of significant interest due to having higher hydrogen permeability at lower temperatures [7]. However, the high cost and low availability of metals such as Pd, Ta, and Nb significantly limits their application as hydrogen-storage alloys. The significantly lower cost of using V alloys in hydrogen purification technologies highlights the attraction to them. In late 2022, the cost of V (as ferrovandium) of $32 USD/kg, was approximately 0.05% of the cost of Pd, which was around $75,000 USD/kg [8]. High purity vanadium fetches a higher price of up to $1000 USD/kg depending on market and purity requirements; however, alloying with much cheaper Al and volume production may reduce this cost.

However, vanadium has the disadvantage of low surface catalytic activity for H_2_ dissociation. In addition, the surface oxides that form readily on the native metal inhibit H_2_ permeation and reduce the H_2_-dissociative activity of the surface [9]. For this reason, several surface modifications have been employed, including thin coatings of Pd, resulting in core-shell like materials [10], and oxide modifications that improve permeability [9].

In this paper, we aim to develop an understanding of the potential technical advantages and disadvantages of core-shell-like-plated, critical-temperature-suppressed alloys for hydrogen storage by assessing the technical merits of the materials. We aim to determine the value of pursuing experimental works on these materials. We also aim to develop a clear understanding of gaps in knowledge that further research should address. We do this by comparing the known technical aspects of the unmodified hydrogen-storage alloy LaNi_5_, LaNi_5_ modified by doping in a polymer matrix, a modified Pd_80_Ag_20_ coated on the published V_90_Al_10_ alloy with a suppressed critical temperature, and pure Pd coated on pure V. The purpose of comparing these four potential combinations of materials was to determine if the modified alloys are likely to present advantages over the unmodified metals that would be more practical to produce.

## 2. Results and Discussion

### 2.1. Description of Materials for Comparison

#### 2.1.1. LaNi_5_ Powder

The properties of LaNi_5_ powder taken from the literature are used directly here. There are some variations in the values quoted in the literature for the apparent or effective thermal conductivities and absorption kinetics which are due to the variations in particle sizes and other micro-effects [11]. The properties of LaNi_5_ that has been pelletized or treated further are not included here to enable a comparison to a well-known material. The density of LaNi_5_ ingots is 7.95 g/cm^3^; however, the powder has a much lower effective density due to the packing of fine particles, which is usually in the range of 0.4 to 0.6 g/cm^3^ [12,13].

#### 2.1.2. LaNi_5_-like Alloy Powder in ABS Copolymer

For the mixed LaNi_5_-like alloy (AB_5_) powder and acrylonitrile–butadiene–styrene powder (ABS), thermal and kinetic properties for each material have been investigated separately. The exceptions are those properties quoted by Pentimalli et al. [5] for the composite materials they published. The LaNi_5_-like alloy was purchased from Ergenics and was the patented Hy-Stor^®^ material MmNi_4.5_Al_0.5_, where Mm (mischmetal) is 34% lanthanum, 49% cerium, 13% neodymium, and 4% praseodymium [5]. The composite was then hot pressed into pellets with a diameter of 6 mm, with the composition being 80 wt.% AB_5_ and 20 wt.% ABS [5]. The length of the pellet was not defined by the authors, so this has been assumed to be 10 mm length for further calculations. The density of the AB_5_ alloy was taken from another study which measured the packed powder density to be 4.67 g/cm^3^ with a porosity of 0.43, giving a solid density of 8.192 g/cm^3^ [14]. If the ABS, which has a density of around 1 g/cm^3^, can fill 90% of the packing voids when compressed under 6 kN and above 130 °C, we assume that the proportion of ABS would be near 0.387, with the remaining 0.043 fraction being voids between the particles. Using these estimates, the effective composite density is 3.909 g/cm^3^.

#### 2.1.3. Core-Shell Vanadium–Palladium

For calculations in this work, the bulk vanadium properties were used, which are assumed to be insensitive to particle size unless the particle size becomes powder-like. The palladium coating on vanadium (C-S V-Pd) has been shown to be effective at catalyzing hydrogen permeation at a thickness of 0.5 µm. To enable comparison to the AB_5_/ABS composite, in the calculations we assumed a 6 mm diameter for 10 mm cylindrical pellets with a 0.5 µm thick Pd layer. For each pellet of PcV, the volume of the V per pellet is 0.28 cm^3^, and the Pd volume is 0.0001225 cm^3^, leading to a mass of V per pellet of 1.71 g and 0.00147 g of Pd.

#### 2.1.4. Core-Shell Vanadium–Aluminium Alloy–Palladium–Silver Alloy

As with the C-S V-Pd, the calculations for the core-shell V_90_Al_10_-Pd_80_Ag_20_ (C-S VAl-PdAg), we assumed a 6 mm diameter for 10 mm long cylindrical pellets with a 5 µm thick layer of Pd alloy on the surface. The volume of the pellets was the same as that of PcV pellets, though the weight was assumed to be different. Literature on the density of the alloys was also not found, so they were calculated from the quoted lattice parameters of ~3.1 Å in the body-centered cubic *α* phase [6], leading to a density of 5.428 g/cm^3^ and a weight per pellet of 1.519 g. Similarly, the Pd_80_Ag_20_ density was calculated from the lattice parameters, 3.9 Å in the body-centered cubic *α* phase [15], leading to a density of 5.97 g/cm^3^ and a weight per pellet of 0.0006 g.

#### 2.1.5. Summary of Material Densities

A summary of the densities of the materials considered here is contained in Table 1.

### 2.2. Comparison of Hydrogen Storage Capacities

Each of the four metallic compounds presented here, V, V_90_Al_10_, Pd, and LaNi_5_, absorb hydrogen to form a solid solution which either retains the initial phase of the material or undergoes a phase transition, depending on the temperature and pressure.

Figure 1 shows a summary of the hydrogen contents of the five materials compared in this study at different temperatures and pressures. In most metal hydrides, higher hydrogen absorption will occur at higher pressures and lower temperatures, and this trend can be seen here by the larger circles clustered at the top left of the graph. However, in most cases, the maximum hydrogen content does not change significantly over the range of temperatures examined here; rather, the pressure required to reach that concentration is low. In addition, the reversible hydrogen content will be discussed further; however, the total hydrogen content is required to understand the dynamics of the materials.

At a low pressure of hydrogen, around 0.2 MPa, and at RT, LaNi_5_ experiences a phase transition to LaNi_5_H_6_, starting with the solid solution of hydrogen (the *α* phase), followed by an inhomogeneous phase mixture (the *α* + *β* phase) corresponding to a plateau region or equilibrium pressure, and finally, a high concentration of hydrogen (the *β* phase) [17].

Like LaNi_5_ hydride, palladium hydride has two potential phases, the *α* and *β* phases. The *α* + *β* mixed phase occurs at temperatures below 568 K and at less than 2 MPa of H_2_ pressure. Palladium silver alloys have been proposed, as they have suppressed *α* to *β* miscibility gaps [18]. The miscibility gap refers to the area in a phase diagram of a multi-component material where two phases can coexist. In the case of PdAg alloys, there are temperature and pressure conditions where both the *α* and *β* exist, and suppressing these conditions leads to less decrepitation of the material. In addition, the Pd_80_Ag_20_ has higher permeability for hydrogen, which represents an advantage for improving hydrogen storage performance [18].

Pure vanadium also has phase transitions in the temperature range considered here—there being a change from *α* to *α* + *β*_1_ to *β*_1_ + *β*_2_ to *β*_2_ + *γ* to pure *γ* phase at the maximum hydrogen concentration, which is near 4 wt.% [19]. The high stability of the *β* phase, however, means that the practical reversable hydrogen storage quantity is closer to 2 wt.% [19].

V_90_Al_10_ does not have a phase change in the temperature range considered here; hence, it should show better reversibility and increased stability of the particles compared to pure hydride V.

**Figure 1 molecules-28-03024-f001:**
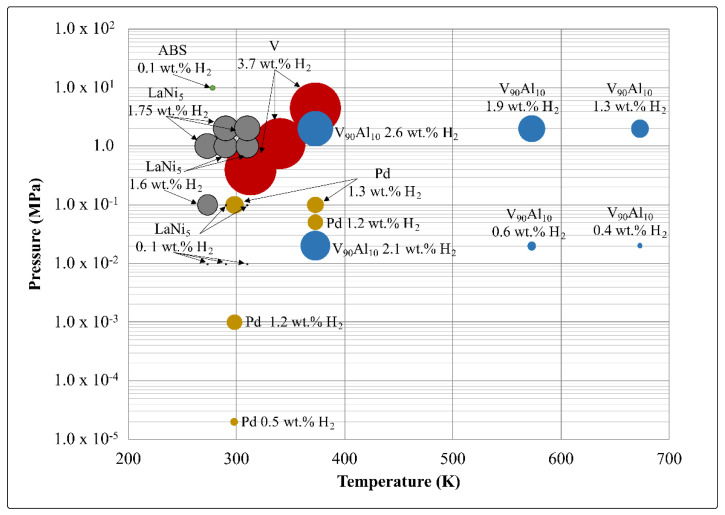
Hydrogen uptake in wt.% in the materials at different temperatures and pressures. The size of the circle corresponds with the gravimetric hydrogen capacity; elements are as labelled [6,20,21,22,23].

### 2.3. Comparison of Hydrogen Storage Properties

There are several properties that are of particular interest for materials for hydrogen storage, in addition to the hydrogen absorption capacity. These include the reversible hydrogen storage capacity, the hydrogen permeation rate, and the physical expansion and contraction of the material during hydriding and dehydriding.

#### 2.3.1. Reversible Hydrogen Storage Capacity

As discussed briefly above, the quantity of hydrogen dissolved in a material is determined by the material’s characteristics, including the thermodynamics of the reaction, and by the temperature and partial pressure of the hydrogen in gas phase around the material. LaNi_5_’s reversible storage capacity reduces by more than 60% over continuous cycling at temperature, which limits practical applications [24]. This reduction in storage capacity is due to disproportionation of the alloy and the formation of a highly stable La-hydride phase that is not reversible under standard cycling conditions [25]. This is the reason for the development of AB_5_-type alloys that do not form the stable La-hydride phase, such as the one used for the AB_5_/ABS experiments which demonstrated increased cyclability [5].

Pure V has even less reversibility. The *β* phase is formed (VH/V_2_H) with the initial introduction of hydrogen even at low pressures. This *β* phase has very high stability and requires the use of vacuum pumps to near 0.1 Pa to reduce the V back to the *α* phase [21]. The phase change is suppressed in V_90_Al_10_ materials to well below room temperature to above 0.8 H/M [6]. This should lead to very high reversibility of hydrogen absorption of near 1.65 wt.%; however, no data for hydrogen desorption was found for this alloy in the literature.

Palladium also undergoes a phase change with hydrogen introduction that can be irreversible if the palladium has a high number of defect sites, consistent with being a highly disordered powder [26]. In most metallic states, Pd has a hydrogen uptake of around 1.8 wt.%. The hydrogen uptake of PdAg alloys shows a continuous decline with increasing Ag content. H_2_ uptake reaches zero at ~60–70% Ag [15]. No discussion of the irreversibility of hydrogen storage in Pd was found in the literature, so we have assumed the reversible hydrogen content per Pd is the same per Pd atom (1.8 wt.%), and no uptake is associated with Ag. This leads to an uptake of 1.35 wt.%, accounting for the reduced Pd content and associated hydrogen, and the addition of Ag atoms with no associated hydrogen. A summary of the reversible hydrogen storage capacity of each of the materials considered in this study is found below in Table 2.

With this information, we can compare the reversible hydrogen storage potential for the pelletized composites discussed here. Figure 2 shows the volume and weight of pellets of each material needed to store 1 kg of H_2_, along with the density of liquid hydrogen for easier comparison to other methods of hydrogen storage [28]. It can be seen here that the C-S V_90_-Al_10_-Pd_80_Ag_20_ composite has higher reversible hydrogen storage density by weight and volume compared to all the other materials presented here.

#### 2.3.2. Hydrogen Diffusion Coefficient

The rate of diffusion of hydrogen is one of the key factors that determine the time taken to hydride and dehydride a material, and this becomes increasingly important as the particle size and packed bed volume increase [30]. Hydrogen diffusion in metals and metal alloys is usually high due to the formation of a so-called ‘solid solution’, where the H_2_ disassociates to protons which diffuse between different sites in the metal crystal lattice. The absolute quantity of hydrogen that can be transported through a material is the permeation rate. The permeation of hydrogen through a material is a product of the solubility and diffusivity of hydrogen in the material. The permeability, quoted in SI units of mol H_2_ m^−1^ s^−1^ Pa^−1^ is strongly dependent on the partial pressure differential of hydrogen in the materials. The solubility of hydrogen determines the number of hydrogen atoms that can be absorbed by the material, which determines the reversible and irreversible hydrogen storage capacity discussed above. The diffusion coefficient describes the ease with which protons move through the material, which considers the activation energy for diffusion. Figure 3 shows the diffusion coefficients in cm^2^/s of each of the materials discussed. What can be seen here is that vanadium has a much higher hydrogen diffusion coefficient than all the other materials, and the V_90_Al_10_ alloy has the second-highest diffusivity. Palladium has different diffusivities depending on the phase. LaNi_5_ and ABS have similar diffusivities. The diffusivity of the exact AB_5_ used in the AB_5_/ABS study was not found in the literature.

#### 2.3.3. Physical Expansion and Contraction

The physical expansion and contraction of metal hydrides and hydride materials derives from two sources, the thermal expansion of the materials and the physical expansion of the lattice because of the introduction of hydrogen into the crystal structure.

As can be seen in Table 3, the thermal expansion values of the materials are significantly less than the physical expansion values of the lattice due to hydrogen absorption. Vanadium has the highest expansion value due to the formation of the *β* phase at the temperatures and pressures considered. The V_90_Al_10_ alloy has a significantly reduced lattice expansion value, more similar to those of Pd and the Pd_80_Ag_20_ alloy. In contrast, the AB_5_ alloy had a larger lattice expansion than pure LaNi_5_.

It is known that the thermal expansion and lattice expansion of materials produce internal strain in them and cause the decrepitation and pulverization of alloys [3]. Figure 4 shows an illustration of the processes that occur in the four pellets discussed here before, during, and after hydrogenation. This illustration is to demonstrate the differences in cycling behavior between the materials discussed in this article. It shows how the LaNi_5_ is pulverized by decrepitation as a single material pellet [24], whereas the ABS protects the AB_5_ alloy from pulverizing [5]. It also illustrates how the V-Pd core-shell material can crack due to the same issue of decrepitation within the VH_x_ solid solution [6], whereas the VAl alloy should not be affected [6].

### 2.4. Comparison by Kinetics of Hydrogen Absorption and Desorption

The kinetics of hydrogen absorption and desorption by LaNi_5_ and similar alloys in powder form are extensively presented and discussed in the literature, and as noted above, do not perform well compared to the composite materials presented. As can be seen, there are several values missing from Table 3 and Table 4 that are required in order to complete full computational dynamics models of the potential for hydrogen storage in these materials. However, from the data available, we can see that the V_90_Al_10_ likely has good thermal conductivity and low hydrogen absorption energy in comparison with the proposed AB5 and AB5/ABS composites [5].

The ideal materials for hydrogen storage by thermal/pressure cycling are those that have high thermal conductivity, low specific heats, and good hydrogen absorption energies [43]. A low absorption energy means the partial pressure of hydrogen will be low in the material, whereas a very high absorption energy means the hydrogen will be difficult to remove from the solid solution, requiring high temperatures. From the data that are available, and presented in Table 4, LaNi_5_ powder is shown to have very poor effective thermal conductivity; however, it is similar to those of the AB_5_ + ABS materials. The low thermal conductivity and higher specific heat of ABS means it is not an ideal material for using as a filler for the hydrogen storage bed, as a significant portion of the heat put into the bed will be used to heat the ABS for no hydrogen desorption. The V_90_Al_10_ alloy does not have as good a thermal conductivity as V; however, the higher thermal conductivity of the Pd_80_Ag_20_ alloy coating should allow the transfer of the heat around the surface of the pellet and result in more uniform dehydrogenation. This could be shown with computational dynamics modelling once further properties of the materials are known.

**Table 4 molecules-28-03024-t004:** The kinetic properties of the materials presented in this study. Room-temperature values are used, except for Pd_90_Ag_10_.

Material	Thermal Conductivity, w/m·K [ref]	Specific Heat, J/g·K [ref]	Hydrogen Absorption Energy, Average or (1st/2nd) kJ/mol H [ref]	Hydrogen Desorption Energy, Total or (1st/2nd) kJ/mol H [ref]	Phase Transition Losses, kJ/mol
LaNi_5_	0.72 [44]		53.2 [44,45]		
LaNi_5_H_6_	0.98 [44]	0.35 [46]			
AB_5_	1.2 [14]		45.5 [47]		
ABS	0.25 [48]	1.9 [38]			
AB_5_/ABS			48.6 [5]		
V	40 [49]	0.71 [50]	(34/40.6) [51]	(20/157) [20]	
Pd	68.8 [52]	0.24	19.1 [53]	18.7 [53]	0.95 [54]
V_90_Al_10_	12 (Estimate from) [55]		29.6 [56]		
Pd_80_Ag_20_	27.4 [52]		7.81 (Pd_90_Ag_10_) * [57]		

* 573.15 K.

## 3. Methods

The physical properties of four materials in the context of hydrogen storage are compared: LaNi_5_, LaNi_5_-like alloy (AB_5_) and acrylonitrile-butadiene-styrene (ABS) copolymer mixtures, core-shell V-Pd, and core-shell V_90_Al_10_-Pd_80_Ag_20_.

To enable this comparison, the specifics of each of the materials were compared while assuming they are produced as pellets like the AB_5_/ABS mixture. This was due to significant unknowns about the properties of each component in the AB_5_/ABS mixture which did not allow disentangling the behavior of the materials to an absolute scale. The other materials discussed here do have intrinsic properties, such as bulk permeabilities, which are available for comparison. For this reason, we completed calculations of the materials as cylinder-shaped pellets 10 mm in length and 6 mm in diameter.

To determine if there is any benefit to further research on these materials, we have calculated the relative densities, calculated reversible hydrogen storage per pellet and at maximum packing densities for cylinders, and reported the thermal and physical expansion, hydrogen permeabilities, and hydrogen absorption kinetics. Modelling of the system using dynamic systems software was not carried out due to a lack of knowledge of the materials discussed. Software used for the calculations here was Microsoft Excel 365 and the open-source Python 3 packages through Anaconda3.

Properties reported here were either taken from the literature directly, which is cited, or calculated using established methods. Calculated densities were derived from the known crystal structure and d-spacing at room temperature using Equation (1).
(1)$$\rho_{x-ray}=\frac{z\times M}{N_{A}\times V}$$
where $\rho_{x-ray}$ is the theoretical density from X-ray data, *z* is the average molecular weight per unit cell, *M* is the number of molecules per unit cell, *N_A_* is Avogadro’s number, and *V* is the volume of the unit cell derived from the X-ray diffraction data.

The density of the materials combined with the volume of the pellets enable the calculation of the weight per pellet using Equation (2).
(2)W=ρ×V
where *W* is the weight of the pellet in each material, ρ is the density of the material, and *V* is the volume of the pellet. For ease of understanding, g/cm^3^ are the units used in this work.

The effective density of the materials was either taken from the literature or calculated using an average of the two materials densities based on the proportion of each. For example, the AB_5_/ABS mixture was quoted as having an 80:20 weight ratio, resulting in a 65:35 volume ratio [5]; hence, the effective density of the pellet was calculated to be proportional to the contribution of each material.

## 4. Conclusions and Recommendations

This desktop technical evaluation of Pd_80_Ag_20_-coated V_90_Al_10_ for the purpose of hydrogen storage has shown that it has excellent potential. V_90_Al_10_ has higher reversible hydrogen-storage content than the other materials considered here, though does not have the highest of all metal hydrides. The combination of good reversible hydrogen-storage content and the use of lighter alloying metals increased the gravimetric hydrogen storage by 1.4 times compared to V/Pd composites. The Pd_80_Ag_20_ and V_90_Al_10_ alloys both also retained the good hydrogen permeabilities, thermal conductivities, and specific heats of the pure Pd and V, respectively.

The main advantage of the V_90_Al_10_ is in the much lower lattice expansion compared to V. In addition, the increased stability of Pd_80_Ag_20_ also represents an improvement on Pd. Core-shell materials made from V_90_Al_10_-Pd_80_Ag_20_ that we discussed here have the potential to reversibly store up to 1.65 wt.% of hydrogen, and provide advantages such as no pulverizing of the hydride bed and stability in air for transport and safe maintenance of a system.

The discussion here focused mainly on cylindrical pellets; however, the high permeability and performance of the vanadium storage materials could allow different shapes and sizes, which could decrease the amount of Pd required to achieve similar hydrogen storage quantities.

Recommendations for future work are experimental determination of the materials’ characteristics to complete the tables above, particularly for information relating to the V_90_Al_10_ and Pd_80_Ag_20_ alloys. This will then allow computation kinetic studies to determine a path forward in terms of size and shape for materials, to take into account thermal and lattice expansion and increase the packing density and accessible surfaces for heat transfer.

## Figures and Tables

**Figure 2 molecules-28-03024-f002:**
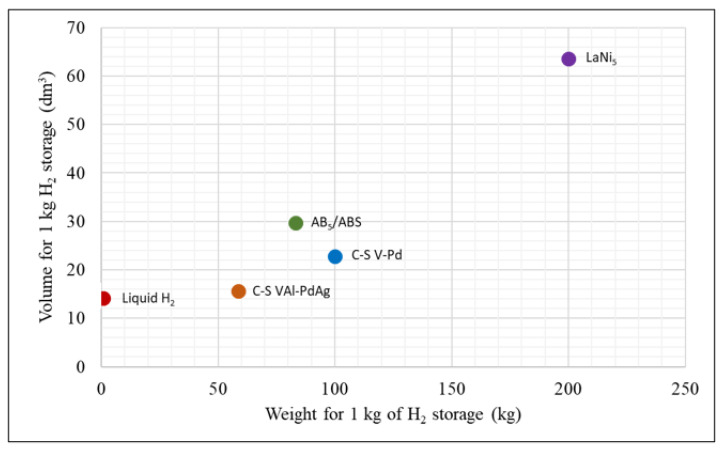
Reversible hydrogen storage of the pelleted materials, except LaNi_5_, which is a powder, as the total weight and volume to store 1 kg of hydrogen at the maximum disordered packing fraction of 0.72 [29].

**Figure 3 molecules-28-03024-f003:**
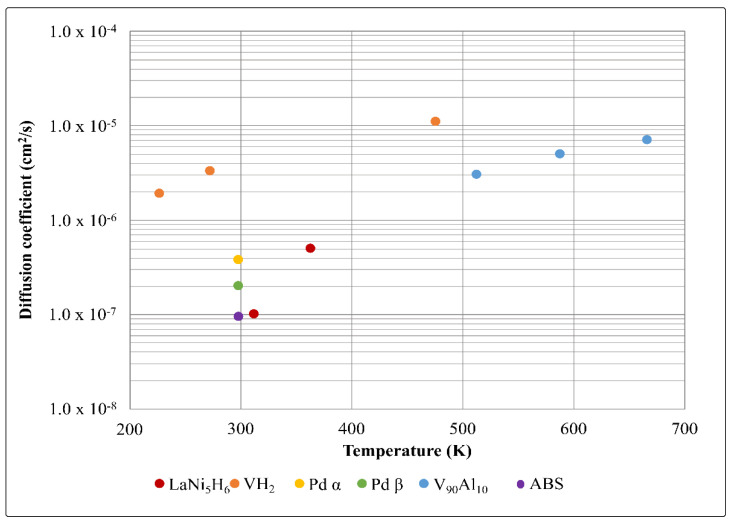
Diffusion coefficients for the materials compared in this study. Coefficient data taken from [19,31,32,33,34,35].

**Figure 4 molecules-28-03024-f004:**
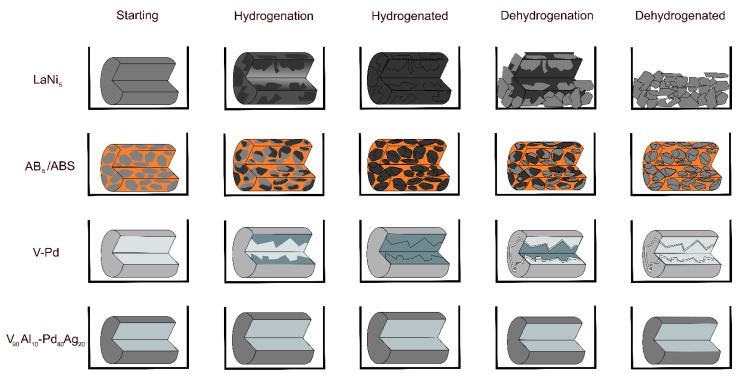
Illustrations of the physical dynamics of hydrogenation and dehydrogenation of the pellets discussed in this article. Within the pellets, the darker color indicates a greater amount of hydrogenation. Orange represents for the AB_5_/ABS the ABS, and the blueish color represents the internal core of V or V_90_Al_10_. The grey outside for the core-shell pellets represents the Pd or Pd_80_Ag_20_ external shell.

**Table 1 molecules-28-03024-t001:** Bulk and effective densities of the materials used in this study.

Materials	Densities—First Material/Second Material (Effective Density) g/cm^3^	First Material Weight per Pellet (g)	Second Material Weight per Pellet (g)	Reference
LaNi_5_	7.95 (4.37)	1.24	N/A	[11]
AB_5_/ABS	8.91/1.03 (3.91 *)	2.52	0.11	[5,14]
C-S V-Pd	6.1/12 (6.10 *)	1.72	0.0015	[16]
C-S VAl-PdAg	5.43/5.97 (5.24 *)	1.53	0.006	[15]

* Calculated values.

**Table 2 molecules-28-03024-t002:** Reversible hydrogen storage for each material considered in this study.

Material	Reversible Storage Capacity in wt.%	Reference
LaNi_5_	0.55	[24]
AB_5_/ABS	1.2	[5]
V	1	[19]
Pd	1.8	[27]
V_90_Al_10_	1.65	Calculated from information in [6]
Pd_80_Ag_20_	1.35	Calculated from information in [15]

**Table 3 molecules-28-03024-t003:** The thermal and hydrogen-driven expansion of the compared materials. The expansion over a temperature range of 200 K is represented in percent change to highlight the difference between thermal and hydrogen-driven expansion. The lattice expansions are compared at room temperature.

Material	Thermal Expansion, K^−1^	Expansion over 200 K, %	Thermal Expansion Reference	Lattice Expansion with Full Hydrogenation, %	Lattice Expansion Reference
LaNi_5_	4.5 × 10^−5^	0.90	[36]	7.4	[17]
AB_5_				16	[37]
ABS	8.2 × 10^−5^	1.64	[38]	None	
V	8.4 × 10^−6^	0.17	[39]	37.7 (phase change)	[40]
Pd	1.18 × 10^−5^	0.24	[41]	2.55	[42]
V_90_Al_10_	Not found			3.278	[6]
Pd_80_Ag_20_	Not found			2.651	[15]

## Data Availability

The data presented in this study are available in the article.

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
