# Peer review of "Core-Shell, Critical-Temperature-Suppressed V Alloy-Pd Alloy Hydrides for Hydrogen Storage—A Technical Evaluation"

_molecules, 2023, doi:10.3390/molecules28073024_

Round 1

Reviewer 1 Report

The article "Palladium coated critical-temperature suppressed vanadium alloy hydrides for hydrogen storage – a technical evaluation" presents an analytical review and comparative analysis of the properties of four hydride materials used for hydrogen storage. Based on a detailed analysis of the literature, the authors collected data regarding such important properties from a practical point of view as bulk and effective density, thermal conductivity, hydrogen adsorption/desorption energies, reversible storage capacity, thermal expansion and lattice expansion during hydrogenation. The collected data can be used as input data for experimental verification of the proposed assumptions or to develop algorithms for computer prediction of relevant properties using computational intelligence tools.

Remarks:

1. In the Results and discussion section, the authors sometimes use the wording "The length of the pellet was not defined by the authors, so this has been assumed to be 10 mm length for further calculations." (lines 114 and 115), "For calculations in this work ..." (line 123). However, in the Methods section, there needs to be more information about the approaches used to carry out the relevant calculations. Additionally, according to the Instructions for Authors (https://www.mdpi.com/journal/molecules/instructions): "Article: These are original research manuscripts. The work should report scientifically sound experiments and provide a substantial amount of new information…." This aspect requires additional argumentation since, in the presented form, the manuscript can be classified as a Review but not an Article.

2. For a better understanding of the text of the manuscript, it is advisable to use the exact names of the same materials in different parts or to detail the reasons for the difference in their designation: for example, Palladium silver alloy coated vanadium aluminum alloy – Pd80Ag20 coated V90Al10 (line 131); PdAcVA (line 133), V90Al10/Pd80Ag20 (Table 1), V90-Al10 (Figure 1), etc.

3. There is no reference to Table 2 in the text.

Author Response

  1. In the Results and discussion section, the authors sometimes use the wording "The length of the pellet was not defined by the authors, so this has been assumed to be 10 mm length for further calculations." (lines 114 and 115), "For calculationsin this work ..." (line 123). However, in the Methods section, there needs to be more information about the approaches used to carry out the relevant calculations.

Response: The Methods section has been updated to provide additional information about the approaches used.

Additionally, according to the Instructions for Authors (https://www.mdpi.com/journal/molecules/instructions): "Article: These are original research manuscripts. The work should report scientifically sound experiments and provide a substantial amount of new information…." This aspect requires additional argumentation since, in the presented form, the manuscript can be classified as a Review but not an Article.

Response: This is a technical assessment, which is conventionally classified as original research rather than review in many areas of academia. This work is original in evaluating the potential for new materials that were developed for different areas of science to be used as hydrogen storage intermediate. As far as the authors are aware, there is no literature on the use of these kinds of materials for hydrogen storage, hence this presents substantial new information regarding this technology. There is a significant gap in the literature that this article addresses.

  1. For a better understanding of the text of the manuscript, it is advisable to use the exact names of the same materials in different parts or to detail the reasons for the difference in their designation: for example, Palladium silver alloy coated vanadium aluminum alloy – Pd80Ag20 coated V90Al10 (line 131); PdAcVA (line 133), V90Al10/Pd80Ag20 (Table 1), V90-Al10 (Figure 1), etc.?

Response: Designations have been changed to specify the exact names of the materials. 

  1. There is no reference to Table 2 in the text.

A reference to Table 2 has been added to the text.

Author Response

The aim of the work is to assess the possibility of using several selected intermetallic alloys for hydrogen storage on a technical scale in hydride tanks. For this purpose, the authors of the manuscript use data on hydrogen absorption properties published and widely available for these materials in the world scientific literature. Based on these data, the authors carried out calculations of analogous properties of some invented alloyed pellets, which would fill hypothetical hydrogen storage tanks for technical purposes. Unfortunately, they do not confirm their calculations with any experimental research. In this situation, the work does not present any significant data valuable for science or technology which are not already known.

After such a statement, I could end my review at this point, but for educational purposes, I allow myself to point out many substantive and editorial errors that I noticed. Below is a list of my comments. My comments are written in Italic.

Response: This is a technical assessment, which is conventionally classified as original research rather than review in many areas of academia. This work is novel in evaluating the potential for new materials that were developed for different areas of science to be used as hydrogen storage intermediate. As far as the authors are aware, there is no literature on the use of these kinds of materials for hydrogen storage, hence this presents substantial new information regarding this technology. There is a significant gap in the literature that this article addresses.

  1. Title : “Palladium coated critical-temperature suppressed vanadium alloy hydrides for hydrogen storage – a technical evaluation.”

The title of the work cannot contain terms or names whose definitions are missing in the work or are only available in the further text. In the cited work, none of these conditions is met.

There are no undefined names or terms used in the title. “Critical temperature” is a well-known chemistry term in the same sense as “enthalpy” or “acid-base reaction”. The words “suppressed”, “technical” and “evaluation” have their English meanings. To assist readers with less knowledge of these terms, we have added “, which is the temperature where further cooling results in a phase change in the alloy,” after the first use of the term in the introduction.

  1. “In this paper, we aim to develop an understanding of the potential technical advantages and disadvantages of plated critical temperature suppressed alloys for hydrogen storage by conducting a technical feasibility assessment.”

Did not understand the meaning of this phrase

It is not clear whether the reviewer does not understand “plated critical temperature suppressed alloys” or “a technical feasibility assessment”. We have expanded this paragraph to further elaborate on both of these phrases.

  1. “the published V90Al10 alloy with a suppressed critical temperature coated with a modified Pd80Ag20 alloy, as well as pure V coated with Pd.
  • technical viability of V90Al10 with a Pd80Ag20 coating as a hydrogen storage material.
  • Materials made with the ratio of V90Al10: Pd80Ag20 discussed here
  • Palladium silver alloy coated vanadium aluminium alloy – Pd80Ag20 coated V90Al10”

If the ratio of materials was 1:1, which one did he cover? Or is it the other way around?

Response: We appreciate the reviewers’ questions, as we can see how the statement “Materials made with the ratio of V90Al10: Pd80Ag20 discussed here” could be unclear. In addition, we interchange between stating “X coated Y” and “Y coated with X”, which we understand may cause confusion despite being synonymous statements. We have to edit the phasing to call the materials using the phasing ‘core-shell V-Pd’ where it should be clear that the V is the core and the Pd is the shell. We have changed the title of the article to reflect this change also.

  1. “missing information that would enable a computational dynamics model to compare all of the materials.

Recommendations for future works are experimental determination of the material characteristics to complete the tables above, particularly for information relating to the 90Al10 and Pd80Ag20 alloys. This will then allow computation kinetic studies to determine a path forward on size and shape for materials to take into account thermal and lattice expansion as well as increase the packing density and accessible surface”

I very much doubt that any common computational algorithm exists for such differently absorbing hydrogen materials as LaNi5 or V , for instance.

Response: The authors cannot comment on the availability of common computational algorithms. However, kinetics models that allow the calculations of thermal and lattice expansion, as well as surface kinetics are common in industrial process studies. Programs such as ANSYS FLUENT also allow for custom calculations of some properties and would likely produce satisfactory results if all required properties of the materials were known.

  1. “Intermetallic metal and solid solution hydrides”

Intermetallic compounds or alloys and solid solutions of hydrogen in metal should probably be more correct.

This has been corrected by removing the words ‘metal’.

  1. “While there are thousands of potential metals and metal alloys”

perhaps too greatly exaggerated , thousands?

Response: We stand by the statement that there are thousands of potential alloys, given that there are ~70 metals and there was no statement on maximum number of components. It is not an exaggeration that there are thousands of possible alloys that could store hydrogen. Despite this, we have changed this statement to “many” to improve readability.

  1. “Another method proposed is the modification of the metal hydrides to change the critical temperature to enable phase-transition free operation at convenient temperatures. Suppression of the critical temperature for alloys has been of interest in metal hydride research disadvantages of plated critical temperature suppressed alloys”

the critical temperature of what? There is no definition!!

The manuscript has been changed to explicitly state this is the critical temperature of the hydride phase.

  1. In late 2022 the cost of V was approximately 0.005 % the cost of Pd per kg.

Are you sure you counted correctly?

Response: Thank-you for noticing this typographical error, it should be 0.05 %. This has been corrected and additional information added to show the sources of the calculations.

  1. “calculations we assume 6 mm diameter by 10 mm length cylindrical pellets with a 0.5 μm 127 thick Pd layer. For each pellet of PcV, the volume of the V per pellet is 0.28 cm3, while the 128 Pd volume is 0.0001225 cm3, leading to a weight of V per pellet.

What was the reason for adopting such dimensions, and what will happen if someone else adopts other values?

These dimensions were chosen to enable comparison to the AB5/ABS material which was produced using a die-based pellet press. The reason for the size of the pellet used was not given, however this size of pellet is common in industry. Different values will result in changes to the total volume, but likely little change to the total weight. Larger pellets would have less surface area to volume ratio, resulting in less Pd or Pd alloy coating, but the proportion of this is already very small. Computational dynamics modelling of the materials with different pellet size parameters would reveal if there was any advantage or disadvantage to different size pellets in terms of heat flow or hydrogen uptake/release rate. To make it more explicit as to the reason for these values, we have added further explanations to the methods section.

  1. Table 1 Second weight pellet (g) ???

Response: Thank you for noticing this error – the wording has been corrected to add the word ‘material’ after ‘Second’.

  1. “Each of the four metallic compounds presented here, the V, V90Al10, Pd, and LaNi5 147 adsorb hydrogen to form either a solid solution which retains the initial phase of the material or undergoes a phase transition, depending on the temperature and pressure.”

The use of the term adsorption in many workplaces is inappropriate and, moreover, unacceptable. The authors use adsorption and absorption interchangeably as if they were the same phenomenon. Adsorption is the phenomenon of physical bonding of gas molecules on the surface of the body, while absorption is the entry of gases, eg hydrogen, into the interior of the crystal lattice as a result of a chemical reaction between the gas and the metal.

Confusing these two processes is unacceptable.

While we are unable to comment on the use of the term adsorption in workplaces, we thank the reviewer for noticing the typographical errors in this article. We have corrected the two cases which ad- was mistakenly used rather than ab-, and as the main point of the article is the absorption of hydrogen, we have chosen to reduce the references to adsorption to reduce potential for confusion.

  1. Figure 1 shows a summary of the hydrogen contents of the five materials compared in this study at different temperatures and pressures.

This type of presentation is not allowed in scientific publications.

Response: Figure 1 shows the hydrogen capacity of the five different materials as a function of both pressure and temperature. As such it simultaneously demonstrates the pressure and temperature dependence of the hydrogen uptake of the materials, as well as comparing the materials. While it does not give precise values for the uptakes (which are only averages in any event) it provides a valuable visual comparison of these average values, and, as such, adds scientific information in a succinct and concise manner. Replacing this information with a table would require the reader to perform manual comparisons, losing the instant visual comparison. The figure does what every figure in a scientific manuscript should do – it provides scientific information in a concise visual manner, adding to the content and impact of the textual information.

  1. “LaNi5 reversible storage capacity reduces by more than 60 % over continuous cycling which limits practical applications”

This statement is not true, because all other materials are subject to some kind of degradation sooner or later depending on the conditions in which they are used.

Response: We interpret this comment to mean that the specific degradation of LaNi5 is not a barrier to its practical deployment. However, this degradation is one of the fundamental reasons, together with low gravimetric capacity, why there is further research into hydrogen storage and very few companies currently developing products using LaNi5. The fact that other materials also degrade does not change the disadvantages of LaNi5 specific properties. The reviewer’s assertion that it is ‘not true’ that degradation of LaNi5 limits practical application is not supported by the literature.

  1.  “The phase change is suppressed in V90Al10 materials to well below room temperature to above 0.8 H/M [4]. This should lead to very high reversibility of hydrogen adsorption of near 3.7 wt%.”

This is a miscalculation. If for vanadium metal H/M = 0.8 then the concentration in is equal to 1.59 wt% and not 3.7 wt%

Response: We thank the reviewer for checking our calculations. We had assumed that the full 3.7 wt.% uptake that is shown in another article for pure V would apply to the alloy but should have calculated the potential based on the 0.8 H/M as per the article we referenced. We have corrected this in the paper and in subsequent calculations for hydrogen storage capacity.

To sum up, not only for the reasons mentioned above, I do not recommend this work for publication, because it basically does not contribute anything new to the knowledge about the properties and possibilities of storing hydrogen in metals and intermetallic compounds

Response: As discussed above, as far as the authors are aware, there is no literature on the use of these kinds of materials for hydrogen storage, hence this presents substantial new information regarding this technology. This article is a relatively short technical assessment which presents the possible advantages of using hydride phase critical temperature suppressed alloys for hydrogen storage in comparison to other methods that have been deployed, such as polymer coatings/mixtures. We believe that this work, along with other technical evaluations, contributes significantly to knowledge of the possibility of hydrogen storage in metal and intermetallic compounds.

Reviewer 3 Report

Dear authors, this paper on Pd/Ag coated V/Al alloy for hydrogen storage contains a lot of useful information as well as identify gaps in knowledge that experimental work could help fill.

The paper could be enhanced with the addition of some citations and clarifications, as stated in the below comments.

Of particular concern, in this reviewer's opinion, starts at line 204 (below comments #4 and 5) where a misrepresentation of a citation occurs as well as an unjustified assumption where your material of interest's capacity was increased by 0.39wt.%, without mention.

How did this happen? It seems very suspicious that your material of interest would show increases in performance without proof such as experimental data, simulated data, assumption justification or citation.

In my opinion, major clarifications need to be added before this article can be published.

1- Please add references for the paragraph at lines 60 - 67

2 - lines 88 - 90, please explain why you did not consider V with Pd/Ag coating and V/Al with Pd coating.

3 - Figure 1 is confusing. I suggest adding the element name and capacity to the circles in the graph and removing the capacity legend at the top

4 - The assumption made at lines 204 to 208 is wrong. As written in the SECOND PARAGRAPH of the cited reference. The total amount of dissolved hydrogen becomes less as the second element increases. The proper assumption is that an alloy composed of 80% Pd + 20% Al will have a capacity equal to 0.8 that of pure Pd.

5 - How do you justify an assumed capacity of 2.19wt% for your Pd/Ag? You stated earlier that Pd/Ag behaves similarly to Pd. Without proper data or justification, this capacity cannot be assumed to be more than 1.44wt%.

6 - Lines 211 - 214 need to be rewritten to reflect previous comments.

7 - Figure 2: choose different colors, these are not easy to discern

8 - Table 4: "textbook values" is not a proper reference. Choose one, cite it. Also, what is the use of this table? Your alloys of interest have no information available.

9 - Line 257 - 259: citation needed

10 - Line 266 - 270: while the coating may make the dehydrogenation of one particle uniform, it is not trivial that this property would be translated when dealing with a bulk sample, i.e. many particles stacked on top of each other with perimeter contact. Did you simulate this situation?

11 - what does "*573.15 K values." mean in the title for table 5? Is it a footnote? If so, it should be added as a proper footnote.

12 - considering the above comments, the conclusions should be revisited.

Author Response

Dear authors, this paper on Pd/Ag coated V/Al alloy for hydrogen storage contains a lot of useful information as well as identify gaps in knowledge that experimental work could help fill.

We thank the reviewer for their positive comments.

The paper could be enhanced with the addition of some citations and clarifications, as stated in the below comments.

1- Please add references for the paragraph at lines 60 – 67

Response: References have been added.

2 - lines 88 - 90, please explain why you did not consider V with Pd/Ag coating and V/Al with Pd coating.

Response: The reason we chose to only include V with a  Pd coating and VAl with a Pd/Ag coating was to allow a comparison between the pure materials and advanced critical temperature supressed alloys. It may be useful in further calculations and experiments to try the alternative arrangement suggested here. However, the authors feel it would not add to aim of the article, which was to compare current and new materials for hydrogen storage applications. 

3 - Figure 1 is confusing. I suggest adding the element name and capacity to the circles in the graph and removing the capacity legend at the top

Response: We thank the reviewer for these suggestions and have implemented these in Figure 1.

4 - The assumption made at lines 204 to 208 is wrong. As written in the SECOND PARAGRAPH of the cited reference. The total amount of dissolved hydrogen becomes less as the second element increases. The proper assumption is that an alloy composed of 80% Pd + 20% Al will have a capacity equal to 0.8 that of pure Pd.

Response: We thank the reviewer for checking this calculation.  It is the case that the referenced article states clearly that this assumption is incorrect. This was mistake on our part and we appreciate the reviewers time in enabling us to correct this error. We have corrected the uptake to take this into account.

5 - How do you justify an assumed capacity of 2.19wt% for your Pd/Ag? You stated earlier that Pd/Ag behaves similarly to Pd. Without proper data or justification, this capacity cannot be assumed to be more than 1.44wt%.

Response: As per the comment above, the reviewer is correct that PdAg alloys have lower hydrogen uptake than pure Pd. The calculation has been revised to a theoretical 1.35 wt.% based on the cited article and the displacement of Pd by Ag atoms.

6 - Lines 211 - 214 need to be rewritten to reflect previous comments.

Response: These lines have been rewritten to implement the corrections pointed out.

7 - Figure 2: choose different colors, these are not easy to discern

Response: Figure 2 has been changed to different colours to improve readability.

8 - Table 4: "textbook values" is not a proper reference. Choose one, cite it. Also, what is the use of this table? Your alloys of interest have no information available.

Response: Table 4 has been changed to provide proper references, as requested. And the point of the table is to show that there is no information on the alloys of interest. Negative space can highlight the problem better than paragraphs of words.

9 - Line 257 - 259: citation needed

Response: A citation has been added.

10 - Line 266 - 270: while the coating may make the dehydrogenation of one particle uniform, it is not trivial that this property would be translated when dealing with a bulk sample, i.e. many particles stacked on top of each other with perimeter contact. Did you simulate this situation?

Response: No, as discussed in the article, there were too many values missing to enable simulations.

11 - what does "*573.15 K values." mean in the title for table 5? Is it a footnote? If so, it should be added as a proper footnote.

This has been moved to a proper footnote.

12 - considering the above comments, the conclusions should be revisited.

The conclusion has been modified based on these changes.

Reviewer 4 Report

This article by Lamb and Webb describes a technical evaluation of a range of Intermetallic metal and solid solution hydrides that could be used for technological application as hydrogen storage materials. These materials including LaNi5, pure Pd and pure V, where compared against V90Al10 with a Pd80Ag20 coating with various characteristics including volume changes and hydrogen storage capacity. It was determined that the against V90Al10 with a Pd80Ag20 coating has great potential due to the reversibility of the material although it is very expensive. The authors have discussed the pro’s and cons of each material in depth and I suppose there will be a followup paper where the authors put their findings into an experimental simulation.

Overall, I believe this article should be published after minor corrections have been made, most of which are superficial in nature.

1)     Check the subscripts of the molecules throughout the abstract, text and references.

2)     Perhaps add a line or two to the abstract to allow readers some context into the area of hydrogen storage?

3)     I feel the introduction is lacking references. The information provided is of course correct but the readers may want references so that they can do further reading.

4)     Line 174. Please check the wording.

5)     Comparing table 2 and figure 1, I see much of the same data is represented in both. I feel both items are required and offer a great perspective. I noted that different references are used for each. Can the authors comment?

6)     Table 4. “Textbook values” are listed as refs. Can a precise ref be provided?

7)     I was surprised that the delta H of des has not been determined. I thought Griessen, Buckley or Evan Gray would have calculated this for at least Pd? DOI: 10.1038/NMAT4480 may have some info for Pd.

Author Response

This article by Lamb and Webb describes a technical evaluation of a range of Intermetallic metal and solid solution hydrides that could be used for technological application as hydrogen storage materials. These materials including LaNi5, pure Pd and pure V, where compared against V90Al10 with a Pd80Ag20 coating with various characteristics including volume changes and hydrogen storage capacity. It was determined that the against V90Al10 with a Pd80Ag20 coating has great potential due to the reversibility of the material although it is very expensive. The authors have discussed the pro’s and cons of each material in depth and I suppose there will be a followup paper where the authors put their findings into an experimental simulation.

Overall, I believe this article should be published after minor corrections have been made, most of which are superficial in nature.

We thank the reviewer for their positive comments.

1)     Check the subscripts of the molecules throughout the abstract, text and references.

Response: These have been checked and any errors corrected.

2)     Perhaps add a line or two to the abstract to allow readers some context into the area of hydrogen storage?

The Abstract has been expanded to include some context into the area of hydrogen storage.

3)     I feel the introduction is lacking references. The information provided is of course correct but the readers may want references so that they can do further reading.

Response: Additional references have been added to the Introduction.

4)     Line 174. Please check the wording.

Response: This has been modified.

5)     Comparing table 2 and figure 1, I see much of the same data is represented in both. I feel both items are required and offer a great perspective. I noted that different references are used for each. Can the authors comment?

Response: Thank you for the confirmation that both Figure 1 and Table 2 are necessary and provide perspective. The different references arise because Figure 1 shows the hydrogen content at various pressures, while Table 2 shows the maximum reversible hydrogen storage potential of the material. These are not always the same values or found in the same articles.

6)     Table 4. “Textbook values” are listed as refs. Can a precise ref be provided?

Response: References have been provided

7)     I was surprised that the delta H of des has not been determined. I thought Griessen, Buckley or Evan Gray would have calculated this for at least Pd? DOI: 10.1038/NMAT4480 may have some info for Pd.

Response: Thank you for noticing this was available and providing directions, the desorption energy was available in the provided reference.

Round 2

Reviewer 2 Report

The revised version of your manuscript makes a much better impression and can be published with a few more comments from me:

1.       Citations

The order of the first citations of individual literature items in the text of the work must agree with the order of the literature items in the bibliography. For example, reference [55] cannot be cited for the first time in the text after reference no [2], as it is in your manuscript. This remark applies to all other quotes, e.g. reference [12] was included in the text between references [3] and [4].

When adding any new literature items in the revised version of the manuscript, you must renumber all references so that the rule of citation order is maintained.

2.        Figures

Don't forget to remove figures 1 and 2 which left from the previous version of the  manuscript.

3.       lines 257, 326, 327, 24

correct the term - adsorption to absorption

Lamba on Lamb

Author Response

1.) Thank you for this comment, we had requested information from the editors in how to modify the bibliography as the version we had we were unable to edit the citations and needed to add the new citations manually at the end of the bibliography. We are aware this requires correction and will ensure it is correct for the published version.

2.) Thank you for this comment, however, we are unable to see the previous versions in the manuscript, only the new versions. We will ensure this is correct in the published version.

3.) Thank you for this comment, we have corrected ad- to ab- sorption. And also the addition of the a on the end of Lamb. We will also conduct a thorough edit of the document before approving the article for publication.

Reviewer 3 Report

Dear authors,

The response to the review comments are appreciated and answer each pointed issue. The modifications made to this paper make it much stronger and clearer. 

I only have a few minor comments:

1. Line 231-232: how can a storage material be more reversible and more stable at the same time? If the hydride is easier to cycle (i.e. more reversible) then it should be less stable. More details are required about the idea being communicated here. 

2. Line 276: is there a more recent source for this information? The cited paper is 59 years old, which is not necessarily an issue but it would be good to have something more recent. 

3. Table 3: Why is the reversible capacity shown in the very confusing "grams of reversible hydrogen stored per pellet" when the rest of the paper talks about weight percentage? This reversible capacity should be changed to weight percent for consistency and clarity.

4. Figure 3 looks like a sketch made in Microsoft Paint. Is this based on data? How do you support this sketch? For such a figure, I would expect SEM data. 

5. A reference needs to be added to the statement at lines 332-334 "from the data available".

Author Response

The authors truly appreciate the reviewers’ time and effort in thoroughly reviewing our paper. This undoubtedly contributed to improving this article. 

1.) Thank you for your comment, we have removed the part of the sentence "like the one used for the AB5/ABS experiments which have been developed to have good cyclability”. This improves the clarity of the statements.

2.) Thank you for your comment. We have used a variety of recent references as well as the original one. Many recent papers reference the original from 59 years ago, so we have chosen to directly reference this paper to direct credit for the work. 

3.) Thank you, we have replaced this table with a figure that includes the same information and this makes this information clearer and less confusing for readers. We have also added additional text. 

4.) We included figure 3 to highlight the differences in cycling behaviour of each of the materials. We have added additional sentences to highlight that this figure is an illustration, along with references to the articles where the behaviour of each material is discussed.

5.) We have added the line "(see tables 4 and 5)" to this sentence to direct readers to the appropriate references.